# Bidirectional View based Consistency Regularization for Semi-Supervised Domain Adaptation

**Yuntao Du**                                                                     *duyuntao@smail.nju.edu.cn*
*National Key Laboratory for Novel Software Technology*
*Department of Computer Science and Technology, Nanjing University*

**Juan Jiang**                                                                    *MF20330037@smail.nju.edu.cn*
*National Key Laboratory for Novel Software Technology*
*Department of Computer Science and Technology, Nanjing University*

**Hongtao Luo**                                                                     *htluo@smail.nju.edu.cn*
*National Key Laboratory for Novel Software Technology*
*Department of Computer Science and Technology, Nanjing University*

**Haiyang Yang**                                                                   *hyyang@smail.nju.edu.cn*
*National Key Laboratory for Novel Software Technology*
*Department of Computer Science and Technology, Nanjing University*

**Mingcai Chen**                                                                  *chenmc@smail.nju.edu.cn*
*National Key Laboratory for Novel Software Technology*
*Department of Computer Science and Technology, Nanjing University*

**Chongjun Wang** *                                                                 *chjwang@nju.edu.cn*
*National Key Laboratory for Novel Software Technology,*
*Department of Computer Science and Technology, Nanjing University*

**Reviewed on OpenReview:** *https://openreview.net/forum?id=WVwnccBJLz*

## Abstract

Distinguished from unsupervised domain adaptation (UDA), semi-supervised domain adaptation (SSDA) could access a few labeled target samples during learning additionally. Although achieving remarkable progress, target supervised information is easily overwhelmed by massive source supervised information, as there are many more labeled source samples than those in the target domain. In this work, we propose a novel method BVCR that better utilizes the supervised information by three schemes, i.e., modeling, exploration, and interaction. In the modeling scheme, BVCR models the source supervision and target supervision separately to avoid target supervised information being overwhelmed by source supervised information and better utilize the target supervision. Besides, as both supervised information naturally offer distinct views for the target domain, the exploration scheme performs consistency regularization within each domain to better explore target information with bidirectional views. Moreover, as both views are complementary to each other, the interaction scheme performs consistency regularization across domains to exchange complementary information for better learning. The proposed method is elegantly symmetrical by design and easy to implement. Extensive experiments are conducted, and the results show the effectiveness of the proposed method.

---

*Corresponding author.

## 1 Introduction

Deep neural networks have achieved remarkable progress in many fields when the train and the test samples are drawn from the same distributions Krizhevsky et al. (2012); He et al. (2016). However, existing deep learning methods need many labeled samples to train the model while annotating the samples is difficult and time-costing in some situations Chen et al. (2019). Besides, the model usually suffers from performance degradation when applying a trained model to a different environment because of domain shift Donahue et al. (2014); Pan & Yang (2010). To solve such a challenge, unsupervised domain adaptation (UDA) has been proposed to address this distribution mismatch by transferring knowledge from the labeled source domain to the fully unlabeled target domain. UDA methods obtain a model that generalizes well on the target domain with no labeled examples from the target domain Pan & Yang (2010); Long et al. (2015).

Recently, Semi-Supervised Domain Adaptation (SSDA) Saito et al. (2019); Jiang et al. (2020) has attracted a lot of attention due to its remarkable performance compared with Unsupervised Domain Adaptation (UDA) Pan & Yang (2010); Long et al. (2015). As few labeled samples of the target domain are available in SSDA and these labeled samples could effectively boost the performance by offering more supervised information.

To better utilize labeled target samples and avoid the learned model being biased towards the source domain, a line of methods learn invariant features to reduce inter-domain discrepancy via minimax entropy Saito et al. (2019), joint representations and risks alignment Li et al. (2021a), and bidirectional adversarial training Jiang et al. (2020). Besides, Kim & Kim (2020) further identifies the intra-domain discrepancy problem within the target domain, and solves this problem by adversarial perturbation. To better reduce the domain discrepancy, following methods reduce both the intra-domain and the inter-domain discrepancy for good adaptation by contrastive learning Li et al. (2021c); Singh (2021), sample-to-sample self-distillation Yoon et al. (2021), and cross-domain adaptive clustering Li et al. (2021b). Beyond these works, DECOTA Yang et al. (2021) proposes to decompose the SSDA task into a semi-supervised learning and an unsupervised domain adaptation task.

Although achieving remarkable progress, there are still some issues that are needed to be addressed to further improve the adaptation performance. For example, existing SSDA methods have not adequately utilized the labeled target samples and target supervised information is easily overwhelmed by massive source supervised information. As there are many labeled samples in the source domain but a few labeled samples in the target domain, therefore the learned discriminative features with a single model are biased towards the source domain Yang et al. (2021). To deal with these issues, we aim to emphasize the importance of the target supervised information and better utilize target supervised information together with source supervised information.

Besides, we find the supervised information from two domains are both useful for the predictions on the unlabeled target samples and complementary to each other. The large amount of labeled source samples is useful for learning discriminative features, while there are domain shifts between the source and the target domain. On the contrary, the labeled target samples share similar distribution with unlabeled target samples (with smaller intra-domain shifts), while the number of labeled samples is limited. As we can see, both supervised information naturally offer two complementary views for the target unlabeled target samples. Thus, we aim to explore the supervised information beneficial to the unlabeled target samples from bidirectional views and combine both information for better learning.

In this work, we propose a simple yet effective method **B**idirectional **V**iew based **C**onsistency **R**egularization (**BVCR**) to achieve the above goals. BVCR is composed of three schemes, i.e., modeling, exploration, and interaction. In the **modeling scheme**, to avoid the target supervised information being overwhelmed by source supervised information, we propose to model both supervised information independently. We learn a classification model for each domain with the same but unshared structure such that the supervised information is modeled by the corresponding model in each domain. In the **exploration scheme**, we further explore both supervised information from different views and we resort to consistency regularization to achieve this goal as it is a simple yet effective solution to various label scare problems Sohn et al. (2020). We design intra-domain consistency regularization (Intra-CR), which is applied to the unlabeled target samples and performed within each domain (model). Especially, the consistency regularization within

the source model could help the source model explore the information relevant to the target domain. And the consistency regularization within the target model could help the target model further explore the internal structure within the unlabeled target samples. In the **interaction scheme**, as both views offer complementary information to each other, we design inter-domain consistency regularization (Inter-CR) to encourage information interaction between both views, which is also applied on the unlabeled target samples but is performed across domains (models). The Inter-CR is performed with an alternative supervised direction such that it could not only promote the learning of each model but also achieve bidirectional transfer across domains.

As we can see, BVCR is a symmetric method in the model architecture, the losses, and the importance of information by design. It is easy to implement without complex adversarial training as previous methods Saito et al. (2019); Li et al. (2021b); Kim & Kim (2020). Extensive experiments are conducted, and the results show the effectiveness of the proposed method. To sum up, the main contributions are as follows:

- We propose an SSDA framework based on bidirectional views to better utilize the supervised information via three schemes, i.e., modeling, exploration, and interaction.

  - In the modeling scheme, BVCR independently models both supervised information to avoid the target supervised information being overwhelmed by source supervised information.
  - In the exploration scheme, we propose Intra-CR to explore the information beneficial to unlabeled target samples from both views separately.
  - In the interaction scheme, we propose Inter-CR to exchange complementary information from bidirectional views to promote the learning of each model.

- We conduct extensive experiments on three common datasets. The proposed method obtains better results than baseline methods in these real-world datasets and we also systematically study the components of BVCR.

## 2 Related Work

### 2.1 Unsupervised Domain Adaptation

Existing unsupervised domain adaptation methods are divided into moment matching and adversarial domain adaptation. The goal of the former is to reduce the statistical distribution discrepancy across domains. The widely used statistical measurements include the first-order moment Long et al. (2015), the second-order momentSun et al. (2016), and other statistical measurements Shen et al. (2018); Lee et al. (2019). Adversarial domain adaptation reduces the distribution discrepancy in an adversarial manner. DANN Ganin et al. (2016) introduces a domain discriminator which plays a min-max game with the feature extractor by the domain adversarial loss. MCD Saito et al. (2018b) introduces two classifiers as a discriminator to play a min-max game with the feature extractor. Considering the practical multi-class problem, MDD Zhang et al. (2019) proposes a margin-based theory and a new method based on this theory is proposed. Following methods Cicek & Soatto (2019) adopt the multi-class discriminator, which considers the domain and class information simultaneously. However, recent works show that the well-known UDA approaches may not generalize well in SSDA scenarios Saito et al. (2019) as they could not effectively utilize the supervised information in the target domain.

### 2.2 Semi-Supervised Domain Adaptation

Semi-supervised domain adaptation is a practical scenario where a few labeled samples are available on the target domain. Early methods focus on reducing inter-domain discrepancy. MME Saito et al. (2019) proposes to align the distribution by minimax entropy. BiAT Jiang et al. (2020) aims to generate bidirectional adversarial examples to fill the domain gap. Furthermore, APE Kim & Kim (2020) introduces the intra-domain discrepancy issue within the target domain in the SSDA problem and solves this issue based on adversarial perturbation. The following methods aim to reduce both inter-domain discrepancy and intra-domain discrepancy simultaneously. ECACL Li et al. (2021c) proposes enhanced categorical alignment to

deal with inter-domain discrepancy and consistency alignment to deal with intra-domain discrepancy. CLDA Singh (2021) designs a contrastive learning framework, where class-wise contrastive learning is adopted to reduce the inter-domain gap and instance-level contrastive alignment are used to minimize the intra-domain discrepancy. CDAC Li et al. (2021b) propose adversarial adaptive clustering loss to minimize both discrepancy. Another line of methods is inspired by the recent process of semi-supervised learning. DECOTA Yang et al. (2021) decomposes SSDA into semi-supervised learning (SSL) sub-task and a UDA sub-task, and it adopts a co-training-based framework to learn from each other. MCL Yan et al. (2022) performs consistency learning at multi-level to fully utilize consistency regularization in SSDA.

### 2.3  Semi-Supervised Learning

Deep semi-supervised learning (SSL) methods include pseudo-labeling Berthelot et al. (2019); Iscen et al. (2019) and consistency regularization. In this work, the most related methods are based on consistency regularization, a variant of self-training. It constrains the learned model to make consistent predictions on the same example under variants of noises. Methods falling into this category differ in the level of noise, models for making two predictions, and the distance measurement. For example, VAT Miyato et al. (2019) generates noise in an adversarial manner. Π-Model Laine & Aila (2017) performs Gaussian noise, dropout, etc., to augment images. Mean Teacher Tarvainen & Valpola (2017) instead maintains an Exponential Moving Average (EMA) of the models parameters. Further, Fixmatch Sohn et al. (2020) proposes to augment the sample with weak and strong augmentation to produce two variants of samples. The learning strategy of our work is inspired by Fixmatch, but the goal is different. Fixmatch is used for improving the performance of supervised learning, while the goal of our method is to transfer knowledge from one domain to the other domain.

## 3  Method

In SSDA, we have datasets sampled from two different domains. Let us consider three datasets in this content: a source dataset contains labeled source samples $\mathcal{D}_s = \{(x_i^s, y_i^s)\}_{i=1}^{N_s}$, a labeled target dataset which contains labeled target samples $\mathcal{D}_t = \{(x_i^t, y_i^t)\}_{i=1}^{N_t}$, and an unlabeled target dataset that contains large number of images without any corresponding labels $\mathcal{D}_u = \{(u_i)\}_{i=1}^{N_u}$. Usually, we have $N_t \ll N_u$ and $N_t \ll N_s$. Labels $y_i^s$ and $y_i^t$ of the samples from source and labeled target sets correspond to the same label space, i.e., $\mathcal{Y} = \{1, 2, ..., K\}$. SSDA seeks to learn a classification model using $\mathcal{D}_s$, $\mathcal{D}_t$, and $\mathcal{D}_u$ and evaluate on $\mathcal{D}_u$.

As shown in Figure 1, the proposed method contains two symmetric models, i.e., the source model $F_s$ and the target model $F_t$. Both models are composed of the feature extractor and classifier, i.e., $F_s = f_s \circ g_s$ and $F_t = f_t \circ g_t$. Different from previous methods Saito et al. (2019); Li et al. (2021c), the structures of both models are the same but the parameters are not shared across the domain. In in the modeling scheme (section 3.1), by learning a separate model for each domain on supervised samples, we could avoid target supervised information getting overwhelmed by source supervised information. In the exploration scheme (section 3.2), before exchanging the information from these two models, we use the unlabeled target samples to adapt the models by intra-domain consistency regularization, such that the source model could concentrate more on the domain-shared information (features) and less on source-private information (features), and the target model could learn better representations by utilizing the structural information within the unlabeled target samples. In the interaction scheme (section 3.3), we exchange complementary information for better learning by inter-domain consistency regularization.

Especially, following the principle of the recent successful semi-supervised learning (SSL) algorithm Sohn et al. (2020), this proposed method utilizes image augmentation techniques with a "weak and strong augmentation" strategy to implement consistency regularization. It uses predictions on weakly augmented images, which are relatively more accurate Chen et al. (2022), to correct the predictions on strongly augmented images. In the next subsections, we introduce each component in detail.

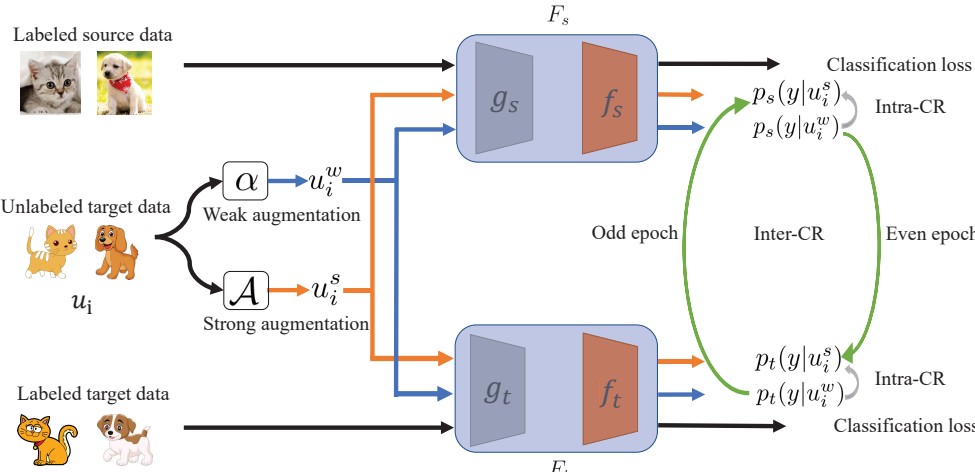

Figure 1: The model structure of BVCR. There are two symmetric models $F_s$ and $F_t$ in BVCR. The Intra-CR is performed within each domain (model) at the first stage, while the Inter-CR is performed across domains (models) at the second stage and it changes the supervised direction every epoch. Note that these two models are trained with different samples, thus, they would not become similar soon.

## 3.1 Modeling: Supervised Training

To model both the source and target supervised information independently, the corresponding models in both domains are trained to minimize the empirical risks on labeled samples as conventional supervised methods. Taking the source model $F_s$ as an example, the feature extractor $g_s$ maps a source sample $x_i^s$ into the feature and the classifier $f_s$ would classify the feature into $K$ categories, i.e., $p_s(y|x_i^s) = F_s(x_i^s) = \sigma(f_s(g_s(x_i^s)))$, where $\sigma$ is the softmax function. Then, cross-entropy loss $\mathcal{L}_{ce}(\cdot, \cdot)$ is adopted to minimize the empirical risk:

$$\mathcal{L}_{src} = \mathbb{E}_{(x_i^s, y_i^s) \sim \mathcal{D}_s} \mathcal{L}_{ce}(y_i^s, p_s(y|x_i^s)) \tag{1}$$

Similarly, the classification loss on the labeled target samples of the target model $F_t$ is,

$$\mathcal{L}_{tar} = \mathbb{E}_{(x_i^t, y_i^t) \sim \mathcal{D}_t} \mathcal{L}_{ce}(y_i^t, p_t(y|x_i^t)) \tag{2}$$

where $p_t(y|x_i^t) = F_t(x_i^t) = \sigma(f_t(g_t(x_i^t)))$ is the prediction on target sample $x_i^t$ of target model $F_t$. For the labeled samples in both domains, besides adopting the cross-entropy loss on the original images, we also apply this loss on both the weakly augmented and strongly augmented images additionally such that the learned model can be more accurate.

## 3.2 Exploration: Intra-domain Consistency Regularization

To better explore the target information from bidirectional views, we introduce intra-domain consistency regularization (Intra-CR) in both domains.

For the target domain, the target model learned with few labeled target samples only contains partial information about the target domain, while the large amount of unlabeled target samples contain more useful information (e.g., internal structure) that is not learned by the model. Thus, performing the consistency regularization within the target domain could help the target model further explore the internal structure within the unlabeled target samples. Besides, by forcing the target model learned with labeled target samples to produce consistent predictions on the unlabeled target samples, the model could achieve implicit alignment between labeled and unlabeled target samples. Thus, it reduces the intra-domain discrepancy within the target domain.

Specially, we follow previous methods Sohn et al. (2020) and adopt the "weak and strong augmentation" strategy to implement consistency regularization. The weak and strong augmentation functions differ in the

degree of image augmentation and more noise is added in strong augmentation. For each unlabeled target sample $u_i$, both the weak augmentation $\alpha(\cdot)$ and strong augmentation $\mathcal{A}(\cdot)$ are applied:

$$u_i^w = \alpha(u_i), \quad u_i^s = \mathcal{A}(u_i) \tag{3}$$

Generally, the predictions on weakly augmented images are more accurate than those on strong augmented images as the latter was more severely damaged Chen et al. (2022). So we use the predictions of weakly augmented images to supervise the predictions on the strong augmentation samples. The pseudo label of the weakly augmented image $\hat{y}_i^{t,w}$ with target model $F_t$ is obtained by,

$$\hat{y}_i^{t,w} = \underset{y \in \{1,\dots,K\}}{\arg\max} \; p_t(y|u_i^w) \tag{4}$$

where $p_t(y|u_i^w) = F_t(u_i^w)$ is the prediction on the weakly augmented sample of target model $F_t$. And the consistency loss within the target domain is,

$$\mathcal{L}_{con}^t = \sum\nolimits_{u_i \in \mathcal{D}_u} [\mathbb{I}(\max(p_t(y|u_i^w)) \geq \gamma)\mathcal{L}_{ce}(\hat{y}_i^{t,w}, p_t(y|u_i^s))] \tag{5}$$

where $\mathbb{I}(\cdot)$ is the indicator function and $p_t(y|u_i^s) = F_t(u_i^s)$ is the prediction on the strongly augmented sample. To mitigate the affect of incorrect pseudo labels, only the samples with high confident predictions are used for loss computation and $\gamma$ is the confidence threshold.

For the source domain, we could simply train source model with only labeled source samples without using any target samples and then apply this model for unlabeled target samples. Such strategy is similar to single-domain domain generalization Zhou et al. (2021a); Wang et al. (2021); Fan et al. (2021). While the results show that the performance of domain generalization methods is worse than that of domain adaptation methods as it is difficult to expect the trained source model to generalize better in the target domain without seeing any target samples Zhou et al. (2021a); Wang et al. (2021).

Thus, we implement the Intra-CR on the source model with unlabeled target samples instead of the source samples. Besides, as the learned source model contains both information specific to the source domain and information shared with the target domain, the Intra-CR within the source domain could help the source model concentrate more on the information relevant to the target domain. Similarly, we also use the predictions of weakly augmented images to supervise the predictions on the strongly augmented samples and the consistency loss within the source domain is,

$$\mathcal{L}_{con}^s = \sum\nolimits_{u_i \in \mathcal{D}_u} [\mathbb{I}(\max(p_s(y|u_i^w)) \geq \gamma)\mathcal{L}_{ce}(\hat{y}_i^{s,w}, p_s(y|u_i^s))] \tag{6}$$

where $p_s(y|u_i^w) = F_s(u_i^w)$, $p_s(y|u_i^s) = F_s(u_i^s)$, and $\hat{y}_i^{s,w} = \arg\max_{y \in \{1,\dots,K\}} p_s(y|u_i^w)$. Combining these two losses, the intra-domain loss is,

$$\mathcal{L}_{con}^{intra} = \mathcal{L}_{con}^s + \mathcal{L}_{con}^t \tag{7}$$

### 3.3 Interaction: Inter-domain Consistency Regularization

Based on the observation that both views are useful for the target domain and are complementary to each other, we propose to make the interaction between them to promote the learning of each model. Thus, we propose inter-domain consistency regularization (Inter-CR), which is applied across models (domains) with unlabeled target samples.

After augmenting original images with weak and strong augmentations, the weakly augmented images are fed to one model and the strongly augmented images are fed to the other model. As the predictions on weakly augmented images are more accurate and we pay equal importance to each model (view), we change the supervised direction every epoch. Namely, we feed the weakly augmented images to the source model in even epoch and feed the weakly augmented images to the target model in odd epoch. The same strategy is also applied to strongly augmented images and note that only the model fed with strongly augmented samples is optimized by Inter-CR.

Taking the latter scenario as an example, the weakly augmented image $u_i^w$ is fed to the target model $F_t$, and we get the pseudo label by Equ 4. The consistency loss across domains at odd epoch is,

$$\mathcal{L}_{con}^{inter} = \sum\nolimits_{u_i \in \mathcal{D}_u} [\mathbb{I}(\max(p_t(y|u_i^w)) \geq \gamma)\mathcal{L}_{ce}(\hat{y}_i^{t,w}, p_s(y|u_i^s))] \tag{8}$$

At even epoch, we change the supervised direction, and the consistency loss across domains is,

$$\mathcal{L}_{con}^{inter} = \sum\nolimits_{u_i \in \mathcal{D}_u} [\mathbb{I}(\max(p_s(y|u_i^w)) \geq \gamma)\mathcal{L}_{ce}(\hat{y}_i^{s,w}, p_t(y|u_i^s))] \tag{9}$$

The proposed Inter-CR has several benefits. Firstly, to some content, the inter-domain consistency regularization can be seen as a variant of co-training Blum & Mitchell (1998). Through this mutual promotion, both models could learn complementary information from each other and be trained better. However, our method is different from the standard co-training framework, as the bidirectional teaching is performed simultaneously and repeated every epoch in co-training. Another strategy to perform Inter-CR is to train the models with simultaneous bidirectional training, i.e., training with Eq 8 and 9 simultaneously every epoch just like standard co-training. Although this strategy is also a possible training method, it may make the two models become similar quickly while BVCR could mitigate this problem by training alternately. Besides, we also experimentally find that alternative teaching achieves better than simultaneous bidirectional training and the comparison is shown in section experiments. Secondly, the Inter-CR can be seen as an implicit alignment across domains, thus reducing the inter-domain discrepancy across domains. Thirdly, previous SSDA methods concentrate on unidirectional knowledge transfer, i.e., from the source domain to the target domain, while the Inter-CR achieves the bidirectional transfer across domains Liu et al. (2021); Xie et al. (2022). Thus, the domain shifts could be better bridged.

### 3.4 Overall

**Strong and Weak Augmentation Strategy.** Data augmentation is an effective way to prevent overfitting and plays an important role in machine learning. In this work, the weak augmentation functions apply random horizontal flip and random crop. RandAugment Cubuk et al. (2020) and Cutout Devries & Taylor (2017) constitute the strong data augmentation. More details about them are reported in the appendix.

**Training Objective.** The overall learning process is composed of two ordinal stages in one epoch and the pseudo code of BVCR is shown in appendix. At the first stage, the objective of our method is a weighted combination of the classification loss in both domains and the intra-domain consistency loss, The optimization problem is formulated as:

$$\min_{g_s,g_t,f_s,f_t} \mathcal{L}_{src} + \mathcal{L}_{tar} + \lambda\mathcal{L}_{con}^{intra} \tag{10}$$

where $\lambda$ is the trade-off hyper-parameter. At the second stage, the learning objective is only the inter-domain consistency loss and the optimization problem at odd (even) epoch is formulated as Equ 11a (Equ 11b):

$$\min_{g_s,f_s} \mathcal{L}_{con}^{inter} \tag{11a}$$

$$\min_{g_t,f_t} \mathcal{L}_{con}^{inter} \tag{11b}$$

### 3.5 Comparison with Existing Methods

In this subsection, we compare BVCR with related methods including Fixmatch Sohn et al. (2020), co-training Blum & Mitchell (1998), and MCL Yan et al. (2022).

- Connection with Fixmatch Sohn et al. (2020): Fixmatch is a representative SSL approach based on consistency regularization. However, such method only implements consistency within one domain. Instead, we follow the same strategy to implement consistency while we not only implement consistency within each domain but also extend it to cross-domain scenario by Inter-CR to effectively exchange the bidirectional supervised information. It is also noticed that although Intra-CR is similar with Fixmatch, while it is the cornerstone of Inter-CR.

- Connection with Co-training Blum & Mitchell (1998): To some degree, the training strategy of Inter-CR could be seen as a variant of co-training. While co-training performs simultaneous bidirectional training but our strategy performs alternative bidirectional training. The proposed strategy could avoid the learned two models becoming similar quickly and the results in section experiments also verifies the superiority of the proposed strategy.

- Comparison with MCL Yan et al. (2022). MCL also adopts consistency regularization for SSDA and it implements consistency at multi level. However, it needs extra class prototypes and complex optimal transport mapping Courty et al. (2017) to perform intra and inter domain consistency. Instead, BVCR adopts unified sample-level consistency, which is more simple.

## 4 Experiments

### 4.1 Setups

**Datasets:** We evaluate the proposed method on several latest SSDA benchmarks including **Office31**[1] Saenko et al. (2010), **Office-Home**[2] Venkateswara et al. (2017), and **DomainNet**[3] Peng et al. (2019). Office31 is a relatively small dataset and it is composed of three domains with 31 classes: Dslr (**D**), Webcam (**W**), and Amazon (**A**). Office-Home is a middle-size dataset, which contains 4 domains including Real (**R**), Clipart (**C**), Art (**A**), and Product (**P**) with 65 classes. DomainNet is the largest benchmark and there are 4 domains with 126 classes. The four domains are Real (**R**), Painting (**P**), Clipart (**C**), and Sketch (**S**).

**Baselines:** We compare **BVCR** with previous methods, including,

- **S+T**, which trains a single model by labeled source and target samples only.

- **SSL methods**: including ENT Grandvalet & Bengio (2004), which trains a single model with labeled source and target and unlabeled target using standard entropy minimization.

- **UDA methods**: including DANN Ganin et al. (2016), ADR Saito et al. (2018a), CDAN Long et al. (2018), and BNM Cui et al. (2020).

- **SSDA methods**: including MME Saito et al. (2019), Meta-MME Li & Hospedales (2020), BiAT Jiang et al. (2020), APE Kim & Kim (2020), UODA Qin et al. (2020), CLDA Singh (2021), DECOTA Yang et al. (2021), and MCL Yan et al. (2022).

**Implementation:** Our method is implemented by PyTorch Paszke et al. (2019), with an open-source library[4] Zhou et al. (2021b). The experiments are conducted on Linux operating system with Tesla V100 (32G memory) and GeForce RTX 3090 (24G memory). We adopt ResNet-34, Alexnet, and VGG-16 pretrained on ImageNet Deng et al. (2009) as the backbone network of the feature extractor. The two classifiers $f_s$ and $f_t$ take a two-layer MLP with randomly initialed weights. All the methods are evaluated under one-shot and three-shot settings, i.e., there are one or three labeled samples in each class. The top-1 accuracy is reported over all the unlabeled samples. At the inference time, we make prediction by equally averaging the source model $F_s$ and target model $F_t$. The results of MCT[5] is implemented by ourselves using the open-source code and the result of baselines are taken from their respective publications if the evaluation protocol is the same.

For the fair comparison, parts of target samples are split as valitaion set, and they are not used for training and only used for selecting the models and hyper-parameters. The split stargey of Office-Home and Domain-Net dataset is following previous work Saito et al. (2019) released on Github[6], while the Office31 dataset is

---

[1]https://www.cc.gatech.edu/~judy/domainadapt/
[2]https://www.hemanthdv.org/officeHomeDataset.html
[3]http://ai.bu.edu/M3SDA/
[4]https://github.com/KaiyangZhou/Dassl.pytorch
[5]https://github.com/chester256/MCL
[6]https://github.com/VisionLearningGroup/SSDA_MME

Table 1: Accuracy (%) on DomainNet under 1-shot and 3-shot settings using Alexnet as backbone.

| Method | R→C | | R→P | | P→C | | C→S | | S→P | | R→S | | P→R | | Mean | |
|---|---|---|---|---|---|---|---|---|---|---|---|---|---|---|---|---|
| | 1-shot | 3-shot | 1-shot | 3-shot | 1-shot | 3-shot | 1-shot | 3-shot | 1-shot | 3-shot | 1-shot | 3-shot | 1-shot | 3-shot | 1-shot | 3-shot |
| S+T | 43.3 | 47.1 | 42.4 | 45.0 | 40.1 | 44.9 | 33.6 | 36.4 | 35.7 | 38.4 | 29.1 | 33.3 | 55.8 | 58.7 | 40.0 | 43.4 |
| DANN | 43.3 | 46.1 | 41.6 | 43.8 | 39.1 | 41.0 | 35.9 | 36.5 | 36.9 | 38.9 | 32.5 | 33.4 | 53.5 | 57.3 | 40.4 | 42.4 |
| ADR | 43.1 | 46.2 | 41.4 | 44.4 | 39.3 | 43.6 | 32.8 | 36.4 | 33.1 | 38.9 | 29.1 | 32.4 | 55.9 | 57.3 | 39.2 | 42.7 |
| CDAN | 46.3 | 46.8 | 45.7 | 45.0 | 38.3 | 42.3 | 27.5 | 29.5 | 30.2 | 33.7 | 28.8 | 31.3 | 56.7 | 58.7 | 39.1 | 41.0 |
| ENT | 37.0 | 45.5 | 35.6 | 42.6 | 26.8 | 40.4 | 18.9 | 31.1 | 15.1 | 29.6 | 18.0 | 29.6 | 52.2 | 60.0 | 29.1 | 39.8 |
| MME | 48.9 | 55.6 | 48.0 | 49.0 | 46.7 | 51.7 | 36.3 | 39.4 | 39.4 | 43.0 | 33.3 | 37.9 | 56.8 | 60.7 | 44.2 | 48.2 |
| Meta-MME | - | 56.4 | - | 50.2 | | 51.9 | - | 39.6 | - | 43.7 | - | 38.7 | - | 60.7 | - | 48.8 |
| BiAT | 54.2 | 58.6 | 49.2 | 50.6 | 44.0 | 52.0 | 37.7 | 41.9 | 39.6 | 42.1 | 37.2 | 42.0 | 56.9 | 58.8 | 45.5 | 49.4 |
| APE | 47.7 | 54.6 | 49.0 | 50.5 | 46.9 | 52.1 | 38.5 | 42.6 | 38.5 | 42.2 | 33.8 | 38.7 | 57.5 | 61.4 | 44.6 | 48.9 |
| CLDA | 56.3 | 59.9 | 56.0 | 57.2 | 50.8 | 54.6 | 42.5 | 47.3 | **46.8** | **51.4** | 38.0 | 42.7 | **64.4** | **67.0** | 50.7 | 54.3 |
| **BVCR** | **61.6** | **62.7** | **56.7** | **58.4** | **54.8** | **56.5** | **47.5** | **50.3** | **46.8** | 49.6 | **51.5** | **51.8** | 61.4 | 62.8 | **54.3** | **56.0** |

Table 2: Accuracy (%) on Office-Home for under 1-shot and 3-shot settings using VGG-16 as backbone.

| Method | R→C | R→P | R→A | P→R | P→C | P→A | A→P | A→C | A→R | C→R | C→A | C→P | Mean |
|---|---|---|---|---|---|---|---|---|---|---|---|---|---|
| | | | | | | 1-shot | | | | | | | |
| S+T | 39.5 | 75.3 | 61.2 | 71.6 | 37.0 | 52.0 | 63.6 | 37.5 | 69.5 | 64.5 | 51.4 | 65.9 | 57.4 |
| DANN | 52.0 | 75.7 | 62.7 | 72.7 | 45.9 | 51.3 | 64.3 | 44.4 | 68.9 | 64.2 | 52.3 | 65.3 | 60.0 |
| ADR | 39.7 | 76.2 | 60.2 | 71.8 | 37.2 | 51.4 | 63.9 | 39.0 | 68.7 | 64.8 | 50.0 | 65.2 | 57.4 |
| CDAN | 43.3 | 75.7 | 60.9 | 69.6 | 37.4 | 44.5 | 67.7 | 39.8 | 64.8 | 58.7 | 41.6 | 66.2 | 55.8 |
| ENT | 23.7 | 77.5 | 64.0 | 74.6 | 21.3 | 44.6 | 66.0 | 22.4 | 70.6 | 62.1 | 25.1 | 67.7 | 51.6 |
| MME | 49.1 | 78.7 | 65.1 | 74.4 | 46.2 | 56.0 | 68.6 | 45.8 | 72.2 | 68.0 | 57.5 | 71.3 | 62.7 |
| BNM | 51.0 | 79.5 | 62.8 | 72.3 | 44.0 | 51.8 | 67.1 | 45.7 | 68.4 | 65.3 | 52.7 | 69.1 | 60.8 |
| UODA | 49.6 | 79.8 | 66.1 | 75.4 | 45.5 | 58.8 | **72.5** | 43.3 | 73.3 | 70.5 | **59.3** | 72.1 | 63.9 |
| DECOTA | 47.2 | 80.3 | 64.6 | 75.5 | 47.2 | 56.6 | 71.1 | 42.5 | 73.1 | **71.0** | 57.8 | 72.9 | 63.3 |
| MCL | 41.4 | 78.9 | 66.1 | **76.6** | 42.2 | **58.6** | 72.2 | 38.8 | 73.6 | 67.5 | 56.5 | 70.2 | 61.9 |
| **BVCR** | **55.6±1.1** | **81.9±0.44** | **68.0±0.52** | 75.7±0.31 | **48.9±1.8** | 57.7±1.1 | 70.2±0.3 | **51.7±1.2** | 72.9±0.4 | 69.4±0.6 | 57.8±0.5 | **73.4±1.4** | **65.3±0.1** |
| | | | | | | 3-shot | | | | | | | |
| S+T | 49.6 | 78.6 | 63.6 | 72.7 | 47.2 | 55.9 | 69.4 | 47.5 | 73.4 | 69.7 | 56.2 | 70.4 | 62.9 |
| DANN | 56.1 | 77.9 | 63.7 | 73.6 | 52.4 | 56.3 | 69.5 | 50.0 | 72.3 | 68.7 | 56.4 | 69.8 | 63.9 |
| ENT | 48.3 | 81.6 | 65.5 | 76.6 | 46.8 | 56.9 | 73.0 | 44.8 | 75.3 | 72.9 | 59.1 | 77.0 | 64.8 |
| APE | 56.0 | 81.0 | 65.2 | 73.7 | 51.4 | 59.3 | 75.0 | 54.4 | 73.7 | 71.4 | 61.7 | 75.1 | 66.5 |
| MME | 56.9 | 82.9 | 65.7 | 76.7 | 53.6 | 59.2 | 75.7 | 54.9 | 75.3 | 72.9 | 61.1 | 76.3 | 67.6 |
| UODA | 57.6 | 83.6 | 67.5 | 77.7 | 54.9 | 61.0 | 77.7 | 55.4 | **76.7** | 73.8 | 61.9 | 78.4 | 68.9 |
| DECOTA | 59.9 | 83.9 | 67.7 | **77.3** | 57.7 | 60.7 | 78.0 | 54.9 | 76.0 | **74.3** | **63.2** | 78.4 | 69.3 |
| MCL | 54.0 | 81.7 | 68.5 | 76.7 | 51.5 | **62.5** | 75.8 | 52.0 | 74.6 | 73.0 | 64.0 | 77.7 | 67.7 |
| **BVCR** | **63.5±0.3** | **84.2±0.14** | **69.1±0.1** | 76.9±0.3 | **60.3±0.5** | 61.8±0.7 | **79.9±0.9** | **58±0.5** | 75.8±0.6 | 73.4±0.1 | 61.7±0.2 | **79±0.6** | **70.3±0.2** |

randomly split by ourself and the split strategy is shown in the code. The specific number of train, validation, and test sets used in three datasets are present in Table 8 in appendix.

Following ReMixMatch Berthelot et al. (2020), we also employ distribution alignment to modify the output of the model for encouraging the unlabeled target distribution of pseudo-labels to follow the source label distribution. We experimentally set the temperature parameter of softmax function as 1 initially and increase it uniformly to 1.5 over the training process to make the predicted probability distribution smoother, avoiding the overconfidence of the model with training.

The learning rate, hyper-parameter $\lambda$, and the initial value of confidence threshold $\gamma$ are searched from {1e-5, 5e-5, 1e-4, 5e-4, 5e-3}, {0.5, 1.0, 2.0}, and {0.5, 0.8, 0.85, 0.9, 0.95} on validation set, respectively. The value of the confidence threshold $\gamma$ is 0.9 and we set $\lambda$ as 1 for all datasets. The number of epoch ans the random seed is all set to be 100 and 123 on three datasets. We set $\lambda$ as 1 for all datasets and batchsize $B$ to be 32, 16, and 16 for Office-Home, Office31, and DomainNet, respectively. We use SGD with momentum for optimization and set the momentum as 0.95. The learning rate is set to be $5e^{-4}$ and adjusted with cosine annealing strategy.

## 4.2 Results

This subsection shows the main results, and more results are shown in the appendix.

**DomainNet.** We summarize the performance of 7 tasks on the DomainNet dataset in Table 1. Using Alexnet backbone, our method outperforms the the second-best method CLDA by 3.6% and 1.7% under 1-shot and 3-shot settings on average. Our approach surpasses the well-known semi-supervised domain adaptation benchmarks methods in most tasks of the DomainNet dataset. Such improved performance

Table 3: Accuracy (%) on Office31 under 1-shot and 3-shot settings with Alexnet.

| Method | W→A | | D→A | | Mean | |
|---|---|---|---|---|---|---|
| | 1-shot | 3-shot | 1-shot | 3-shot | 1-shot | 3-shot |
| S+T | 50.4 | 61.2 | 50.0 | 62.4 | 50.2 | 61.8 |
| DANN | 57.0 | 64.4 | 54.5 | 65.2 | 55.8 | 64.8 |
| ADR | 50.2 | 61.2 | 50.9 | 61.4 | 50.6 | 61.3 |
| CDAN | 50.4 | 60.3 | 48.5 | 61.4 | 49.5 | 60.8 |
| ENT | 50.7 | 64.0 | 50.0 | 66.2 | 50.4 | 65.1 |
| MME | 57.2 | 67.3 | 55.8 | 67.8 | 56.5 | 67.6 |
| BiAT | 57.9 | **68.2** | 54.6 | 68.5 | 56.3 | **68.4** |
| APE | - | 67.6 | - | 69.0 | - | 68.3 |
| **BVCR** | **62.6** | 66.3 | **58.3** | **70.0** | **60.5** | 68.1 |

shows that our approach better utilizes the supervised information in SSDA. In some tasks (e.g., P→R), BVCR does not achieve the best performance. It may be because that the domain discrepancy between task P→R is smaller than other tasks (with better S+T accuracy) and supervised information across domains are less complementary. Thus, BVCR gets less gain in these tasks. But with the proposed Intra-CR and Inter-CR, BVCR also achieves the second-best results.

**Office-Home.** Table 2 reports the results of the Office-Home dataset under 1-shot and 3-shot settings with VGG-16. Our method shows the best performance in most of the scenarios. Specially, the improvement of mean accuracy is 1.4% and 1.0% compared with the second-best method in 1-shot and 3-shot, respectively. Our method achieves more performance gain when giving less labeled target samples. Similar to the results on DomainNet, our approach surpasses the state-of-the-art SSDA approaches in most of the adaptation tasks. Besides, In some domain adaptation cases under 1-shot, such as R→C and A→C, the proposed method exceeds DECOTA and MCL by more than 3%.

**Office31.** The results of the Office31 dataset with Alexnet are shown in Table 3. Our method also outperforms other baselines in most tasks. The improvement is 4.2% compared with BiAT under the 1-shot setting and the results of 3-shot is slightly lower than BiAT by 0.3% on average. Improvement on this dataset is less than that of Office-Home and DomaiNet may be due to small size of samples.

### 4.3 Insight Analysis

**Ablation study.** The proposed method contains three losses (schemes): the classification loss (modeling scheme), the Intra-CR loss (exploration scheme), and the Inter-CR loss (Interaction). We investigate the effect of each component by different combination during training. Besides, we also analyze the effectiveness of data augmentation (both strong and weak augmentation) on labeled samples. The results on the tasks A→C and R→P of Office-Home dataset under 3-shot setting with VGG-16 are shown in Table 4. As we can see, data augmentation is an effective method for improving generalization, and it improves the performance by 5.2% and 2.1% when only using classification losses. Following settings all adopt the data augmentation on labeled samples. As for the losses, after adding the Intra-CR loss, the performance is improved by 3.1% and 2.4%, which implies that the Intra-CR can help the models better explore the target information from

Table 4: Results (%) of ablation study on two tasks of Office-Home under 3-shot.

| Augmentations on labeled samples | $\mathcal{L}_{src}$ | $\mathcal{L}_{tar}$ | $\mathcal{L}_{con}^{intra}$ | $\mathcal{L}_{con}^{inter}$ | A→C | R →P |
|---|---|---|---|---|---|---|
| | √ | √ | | | 42.8 | 77.5 |
| √ | √ | √ | | | 48.0 | 79.6 |
| √ | √ | √ | √ | | 51.1 | 82.0 |
| √ | √ | √ | √ | √ | **57.4** | **84.4** |

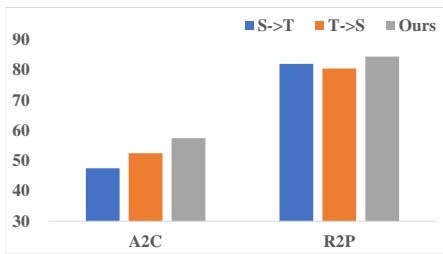 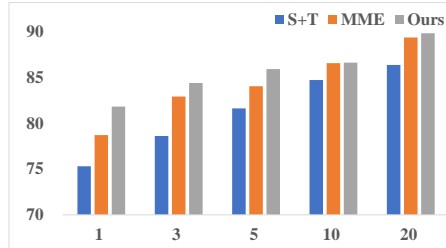

Figure 2: (a) Analysis of supervised direction. (b) Impact of varying labeled samples.

different views. Moreover, by interacting both views with Inter-CR, the performance is further improved by 6.3% and 2.4% and the combination of all losses achieves the best results.

**Effect of backbone.** We analyze the network architecture with Alexnet, VGG-16, and ResNet-34 on Office-Home with four tasks under 3-shot. As shown in Table 5, we can see that our BVCR outperforms all comparison methods on average accuracy and achieve 55.1%, 69.2%, and 73.4% mean accuracy with Alexnet, VGG-16, and ResNet-34, respectively. The results on other tasks are shown in appendix and our method also achieves better performance. The different results with different backbones demonstrate that our method works well with multiple network structures.

Table 5: Accuracy (%) on Office-Home under 3-shot setting using Alexnet, VGG-16, and ResNet-34 as backbone networks.

| Network | Method | R→P | P→A | A→C | C→R | Mean |
|---------|--------|------|------|------|------|------|
| Alexnet | S+T | 66.7 | 36.1 | 38.8 | 54.3 | 49.0 |
| | APE | **74.6** | **42.1** | 44.5 | 58.1 | 54.8 |
| | **BVCR** | 74.0 | 41.9 | **45.3** | **59.4** | **55.1** |
| VGG-16 | S+T | 78.6 | 55.9 | 47.5 | 69.7 | 62.9 |
| | APE | 81.0 | 59.3 | 54.4 | 71.4 | 66.5 |
| | **BVCR** | **84.4** | **61.7** | **57.4** | **73.4** | **69.2** |
| ResNet-34 | S+T | 80.8 | 63.5 | 54.0 | 68.3 | 66.7 |
| | APE | 86.2 | 66.1 | 63.9 | **76.8** | 73.3 |
| | **BVCR** | **86.9** | **66.8** | **64.7** | 75.1 | **73.4** |

**Effectiveness of bidirectional training.** Inter-CR is implemented in an alternative bidirecitional training manner. Another strategy to implement Inter-CR is to perform Inter-CR with Equ 8 and Equ 9 simultaneously like co-training and we denote this strategy as simultaneous bidirectional training. We compare these two strategies on tasks A→C and P→R and the results are shown in Table 6. We can see that alternatively training two models achieves better performance as it could effectively avoid these two models becoming similar quickly and encourage better information exchange.

Table 6: Analysis of training strtegy.

| | A→C | R→P |
|---|------|------|
| Simultaneously | 53.7 | 83.2 |
| Alternatively | **57.4** | **84.4** |

**Analysis of supervised direction in Inter-CR.** To analyze the effect of changing supervised direction in Inter-CR, we compare our method with two variants on tasks A→C and R→P of Office-Home under 3-shot. 1) We always feed the weakly augmented samples to the source model (denoted as S → T). 2) We always feed the weakly augmented samples to the target model (denoted as T → S). Note that the supervised direction of the former is always from the source domain to the target domain, and the direction of the latter is always from the target domain to the source domain. While the supervised direction is changed every epoch in BVCR. The results are shown in Figure 2(a). As we can see, the proposed strategy achieves better performance for both tasks, which implies that bidirectional transfer across domains could yield better adaptation than unidirectional transfer.

**Impact of the number of labeled samples.** We compare the performance changes with the number of labeled target samples from 1 to 20 for each class on task R→P on Office-Home with VGG-16. As shown in

Figure 2(b), the performance of all methods is increased with more labeled target samples. Besides, BVCR consistently reaches the best performance for all the cases. The results show the benefit of our method for flexibly exploiting a different number of labeled target samples.

**Feature visualization.** To qualitatively evaluate the results, we adopt the t-SNE van der Maaten & Hinton (2008) to visualize the features produced by our method and the S+T method for adaptation task A→C on the Office-Home dataset under 3-shot setting. The results in Figure 3(a) and 3(b) show that the learned features exhibit a favorable clustering structure, which implies the proposed methods could achieve cross-domain alignment and learn discriminative features.

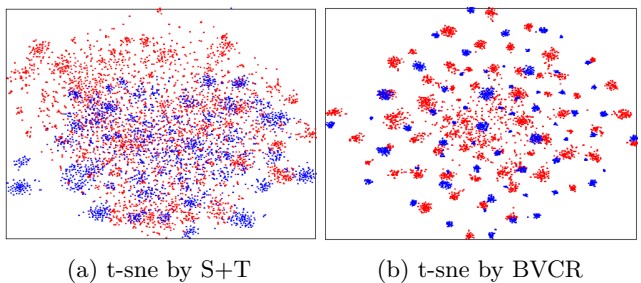

(a) t-sne by S+T       (b) t-sne by BVCR

Figure 3: Feature visualization by t-sne

**Comparison with Fixmatch** We compare with Fixmatch on Office-Home dataset with VGG backbone under 3-shot and the results are shown in Table 7. As we can see, Fixmatch achieves better performance than S+T but the improvement is limited, i.e., 0.5%, which implies that simply treating a semi-supervised domain adaptation(SSDA) problem as a semi-supervised learning (SSL) problem is a trivial solution as this strategy could not deal with the domain distribution shifts effectively. BCVR considers the SSDA problem from a complementary view, i.e., intra-domain supervised information and inter-domain supervised information, which could better deal with the domain discrepancy and promote model learning.

Table 7: Comparison with Fixmatch on Office-Home for under 3-shot settings using VGG-16 as backbone.

| Method | R→C | R→P | R→A | P→R | P→C | P→A | A→P | A→C | A→R | C→R | C→A | C→P | Mean |
|---|---|---|---|---|---|---|---|---|---|---|---|---|---|
| S+T | 49.6 | 78.6 | 63.6 | 72.7 | 47.2 | 55.9 | 69.4 | 47.5 | 73.4 | 69.7 | 56.2 | 70.4 | 62.9 |
| Fixmatch | 52.3 | 78.7 | 63.7 | 72.9 | 47.7 | 54.9 | 70.5 | 49.7 | 72.8 | 69.4 | 57.7 | 70.0 | 63.4 |
| **BVCR** | **63.2** | **84.4** | **69.1** | **76.7** | **60.9** | **61.7** | **79.0** | **57.4** | **76.1** | **73.4** | **62.4** | **79.4** | **70.3** |

## 5 Conclusions

In this paper, we focus on SSDA. Considering that target supervised information is easily overwhelmed by massive source supervised information in existing SSDA methods, we propose BVCR to model the source supervised information and supervised target information separately. As both of the supervised information are complementary to each other, we explore the target information from both views and make information interaction between views. With the help of consistency regularization, we design Intra-CR and Inter-CR to achieve the above goals. The experimental results show that the proposed method outperforms baselines. As for limitation, our method adopts pseudo labels for learning and it may be affected by noisy pseudo labels. Thus, it is our further work to explore a more flexible strategy to generate or filter unreliable pseudo labels than a fixed threshold.

## Acknowledgement

This paper is supported by the National Natural Science Foundation of China (Grant No. 62192783, U1811462), the Collaborative Innovation Center of Novel Software Technology and Industrialization at Nanjing University.

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

Table 8: The number of train set, validation set, and test set when each domain is selected as target domain under 3-shot setting.

|  | Office31 | | | Office-Home | | | | DomainNet | | | |
| --- | --- | --- | --- | --- | --- | --- | --- | --- | --- | --- | --- |
|  | Dslr | Webcam | Amazon | Real | Clipart | Art | Product | Real | Painting | Clipart | Sketch |
| # Labeled target samples in train set | 93 | 93 | 93 | 195 | 195 | 195 | 195 | 378 | 378 | 378 | 378 |
| # Unlabeled target samples in train set | 312 | 609 | 2631 | 3967 | 3975 | 2037 | 4049 | 69602 | 30746 | 17947 | 23448 |
| # Validation | 93 | 93 | 93 | 195 | 195 | 195 | 195 | 378 | 378 | 378 | 378 |
| # Test | 312 | 609 | 2631 | 3967 | 3975 | 2037 | 4049 | 69602 | 30746 | 17947 | 23448 |

# A    Setup

## A.1    Augmentation and pseudo code

In this paper, the weak and strong augmentation functions differ in the degree of image augmentation and more noise is added in strong augmentation. Specifically, the weak image transformation function applies random horizontal flip and random crop. Two augmentation techniques, namely RandAugment Cubuk et al. (2020) and Cutout Devries & Taylor (2017), constitute the strong transformation function. In RandAugment, a given number of operations are randomly selected from a fixed set of geometric and photometric transformations, such as affine transformation, color adjustment. Then they are applied to images with a stochastic magnitude. The operations of RandAugment are shown in Table 10. The range is similar to the original version, so we do not elaborate on their meaning here. Cutout randomly masks out square regions of images to gray. Both of them are applied sequentially in the strong augmentation. Besides, the pseudo code of BVCR is shown in algorithm 1.

# B    More Results

**More results on DomainNet.**    We summarize the performance of 7 tasks on the DomainNet dataset in Table 9. It is obvious that the proposed method outperforms baseline methods on most tasks. On average, our method outperformed the best-performed method APE by 1.2% under the 3-shot setting when ResNet-34 is adopted as the backbone network.

**Analysis of training cost.**    We test the training time of MME, UODA, and BVCR on task A→C of Office-Home dataset under 3-shot with the same running steps. The comparison of running time for different methods is shown in Figure 4. As we can see, BVCR needs more training time, namely 2.2 times of MME and 1.07 times of UODA. It is because that BVCR has two models and the samples are fed into the models some times. But recent SSDA methods also need much training time as they rely on contrastive learning or adversarial training.

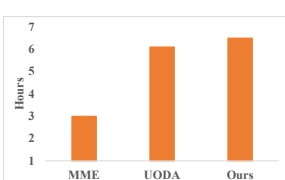

Figure 4: Training cost of different methods

Table 9: Performance comparison in DomainNet under 3-shot setting using ResNet-34 as backbone network.

| Method | R→C | R→P | P→C | C→S | S→P | R→S | P→R | Mean |
| --- | --- | --- | --- | --- | --- | --- | --- | --- |
| S+T | 60.0 | 62.2 | 59.4 | 55.0 | 59.5 | 50.1 | 73.9 | 60.0 |
| DANN | 59.8 | 62.8 | 59.6 | 55.4 | 59.9 | 54.9 | 72.2 | 60.7 |
| ADR | 60.7 | 61.9 | 60.7 | 54.4 | 59.9 | 51.1 | 74.2 | 60.4 |
| CDAN | 69.0 | 67.3 | 68.4 | 57.8 | 65.3 | 59.0 | 78.5 | 66.5 |
| ENT | 71.0 | 69.2 | 71.1 | 60.0 | 62.1 | 61.1 | 78.6 | 67.6 |
| MME | 72.2 | 69.7 | 71.7 | 61.8 | 66.8 | 61.9 | 78.5 | 68.9 |
| UODA | 75.4 | 71.5 | 73.2 | 64.1 | 69.4 | 64.2 | 80.8 | 71.2 |
| Meta-MME | 73.5 | 70.3 | 72.8 | 62.8 | 68.0 | 63.8 | 79.2 | 70.1 |
| BiAT | 74.9 | 68.8 | 74.6 | 61.5 | 67.5 | 62.1 | 78.6 | 69.7 |
| APE | **76.6** | 72.1 | **76.7** | 63.1 | 66.1 | 67.8 | **79.4** | 71.7 |
| **BVCR** | 76.4 | **73.9** | 71.3 | **69.2** | **71.2** | **69.4** | 79.0 | **72.9** |

---

**Algorithm 1** Bidirectional View based Consistency Regularization for SSDA (**BVCR**)

---

**Input:** Source samples $\mathcal{D}_s$, Target labeled samples $\mathcal{D}_t$ , Target unlabeled samples $\mathcal{D}_u$, Batch-size $B$, trade-off hyper-paramter $\lambda$, and running epoches $T$.
**Output:** Trained target model $F_t$ and source model $F_s$.
1: Initialize t = 1
2: **while** $t \leq T$ **do**
3:      // The first stage
4:      Sample a mini-batch of $B$ examples from $\mathcal{D}_s$ and $\mathcal{D}_t$ separately, $B$ examples from $\mathcal{D}_u$.
5:      Train the feature extractors $g_s$, $g_t$ and classifiers $f_s$, $f_t$ by Equ 10.
6:      // The second stage
7:      Sample a mini-batch of $B$ examples from $\mathcal{D}_u$.
8:      **if** odd epoch **then**
9:          Train the source feature extractor $g_s$ and source classifier $f_s$ by $\mathcal{L}_{con}^{inter}$ (Equ 8).
10:     **else**
11:         Train the target feature extractor $g_t$ and target classifier $f_t$ by $\mathcal{L}_{con}^{inter}$ (Equ 9).
12:     **end if**
13: **end while**=0

---

Table 10: List of operations for strong transformations of the RandAugment. Two transformations are randomly chosen and performed with stochastic magnitude.

| Operation | Range |
|---|---|
| Identity | [0.0, 1.0] |
| ShearX | [0.0, 0.3] |
| ShearY | [0.0, 0.3] |
| TranslateX | [0.0, 0.33] |
| TranslateY | [0.0, 0.33] |
| Rotate | [0, 30] |
| AutoContrast | [0, 1] |
| Invert | [0, 1] |
| Equalize | [0, 1] |
| Solarize | [0, 110] |
| Posterize | [4, 8] |
| Color | [0.1, 1.9] |
| Brightness | [0.1, 1.9] |
| Sharpness | [0.1, 1.9] |

