# OpenReview forum: "Bidirectional View based Consistency Regularization for Semi-Supervised Domain Adaptation"
_TMLR — Accepted by TMLR_

### Review · Reviewer_rief · 2022-12-02

**Summary Of Contributions:**

This paper focuses on the problem of semi-supervised domain adaptation (SSDA). The authors propose a new method called BVCR which model the source supervised information and supervised target information separately. BVCR designs Intra-CR and Inter-CR to explore the target information from both views and make information interaction between views.



**Audience:**

Yes

**Broader Impact Concerns:**

No other impact concerns.

**Claims And Evidence:**

Yes

**Requested Changes:**

1.  In section 3.5, author say that BVCR’s unified sample-level consistency is more effective than MCL’s. Further experiment and analysis is needed.

2.  Further analysis and experiment of this result is required. And more experiments in deeper models like ResNet34 and ResNet50 are recommended.

**Strengths And Weaknesses:**

Pros
1 The paper is well-organized.
2 Clear and concise image of overall model structure.
3 Sufficient analysis and ablation study.

Cons
1 Novelty : This paper is like a combination of Fixmatch and co-teaching or a co-teaching version of MCL[1].
2 In section 3.5, author say that BVCR’s unified sample-level consistency is more effective than MCL’s. Further experiment and analysis is needed.
3 Results : The results are less convincing because the authors didn’t report the results of other methods include MCL in Table 3 and Table 8. It seems that MCL surpasses BVCR by up to 3.6% in the setting of dataset DomainNet and backbone ResNet34. Further analysis and experiment of this result is required. And more experiments in deeper models like ResNet34 and ResNet50 are recommended.

Missing Reference
Some papers related to semi-sipervised domain adaptation are missing.

Semi-supervised heterogeneous domain adaptation: Theory and algorithms (TPMAI),
Semi-supervised Domain Adaptation via Minimax Entropy (ICCV),
Frustratingly easy semi-supervised domain adaptation (ACL),



[1] Zizheng Yan, Yushuang Wu, Guanbin Li, Yipeng Qin, Xiaoguang Han, and Shuguang Cui. Multi-level consistency learning for semi-supervised domain adaptation. In IJCAI, 2022.

---

### Review · Reviewer_E8Hz · 2022-12-07

**Summary Of Contributions:**

This paper deals with the problem of semi-supervised domain adaptation (SSDA). The authors propose a method BVCR that utilizes the supervised information by three schemes, i.e., modeling, exploration, and interaction. Specifically, it proposes Intra-CR to explore the information beneficial to unlabeled target samples and Inter-CR to exchange complementary information from bidirectional views to promote the learning of each mode

**Audience:**

Yes

**Claims And Evidence:**

Yes

**Requested Changes:**

1. Clarification on Inter-domain Consistency w.r.t why you claim it could exchange complementary information from bi-views.
2. Recommend adding a Fixmatch baseline
3. Verify whether $L_{Intra-source}$+$L_{Intra-target}$ is better than $L_{Intra-target}$
4. For ablation table, recommend adding  $L_{Inter} + L_{supervised}$.

**Strengths And Weaknesses:**

**Pros**
- The paper is easy to follow and well-organized.
- Experiments on several datasets and analyses are sufficient.

**Cons**
- Limited Novelty: this paper is kind of like a combination of Fixmatch + co-training. Recent works have explored intra-domain consistency[1] and co-training[2]. Even though the authors specify the difference in the main text, I still consider this work as incremental.
- Lack of Justification on Inter-domain Consistency. The source model and target model are trained in the same way at stage 1 Eqn 10. Then, the authors design a co-training loss for inter-domain consistency and claim it could ** exchange complementary information from
bidirectional views**. However, I cannot see how these two models hold complementary information, considering they are trained exactly the same at stage 1. No further analysis or justification for verifying this claim. Also, the authors do not explain why iterative optimization (odd, even epochs) is necessary. Essentially, it is like making the model consistent with its moving average (In this case, the moving average only considers the model from the last epoch).
- Experiment:
1. Intra-domain consistency: authors should at least compare to a Fixmatch baseline and [1], which share large similarity with the proposed component. Also, the authors claim that they also do source domain consistency compared to Fixmatch. But the benefits of it do not reflect in Table 4.  How $L_{Intra-source}$+$L_{Intra-target}$ is better than $L_{Intra-target}$?
2. Inter-domain Consistency: First, I recommend the authors show $L_{Inter} + L_{supervised}$ in Table 4. Although the performance of this paper is better than DECOTA in Table 1-2, it is unclear how the co-training part of this paper compares to DECOTA. Because $L_{Intra}$ plays a very important role in the performance. We cannot see how the proposed Inter-domain Consistency is better than other co-training strategies.

[1] Kai Li, Chang Liu, Handong Zhao, Yulun Zhang, and Yun Raymond Fu. Ecacl: A holistic framework
for semi-supervised domain adaptation. 2021 IEEE/CVF International Conference on Computer Vision
(ICCV)
[2] Luyu Yang, Yan Wang, Mingfei Gao, Abhinav Shrivastava, Kilian Q. Weinberger, Wei-Lun Chao, and SerNam Lim. Deep co-training with task decomposition for semi-supervised domain adaptation. Proceedings
of the IEEE/CVF international conference on computer vision (ICCV), 2021.

---

### Review · Reviewer_m1Ua · 2023-01-11

**Summary Of Contributions:**

Summary: The authors propose an algorithm for semi-supervised domain adaptation using a combination of intra and inter consistency regularization using three different strategies consisting of modeling, exploration and interaction. While semi-supervised domain adaptation has shown immense improvement with the availability of small amount of labeled target data, they are overwhelmed with the large number of labeled source samples. This undermines the importance and usage of labeled target samples in an efficient way. In this paper, the authors propose to have complimentary viewpoints by using two different models for source and target data. In the modeling stage, the source model is trained on the fully labeled source data while the target data is trained on the labeled target data. In the exploration stage, the unlabeled target samples are augmented into weak and strong augmented versions and are passed through both the models. The weaker augmented samples are used as target labels while the stronger augmented sample predictions are used as model predictions. Then the intra domain consistency regularization is performed by both, the source and target models on the unlabeled target samples. The interaction stage involves using the weakly augmented sample predictions of one model and the strongly augmented sample predictions from the second model to perform inter-domain consistency regularization and this utilizes the complimentary viewpoint towards developing a better generalized model.

**Audience:**

Yes

**Broader Impact Concerns:**

No.

**Claims And Evidence:**

Yes

**Requested Changes:**

Addressing the above weakness points of [1] and [2] is critical for acceptance of the paper while the points [3] and [4] would make the paper stronger with better convincing evidence for the algorithm.

**Strengths And Weaknesses:**

Strengths:
1. Well written paper and has the required flow which explains the concept with ease.
2. Good ablation studies on the analysis of the supervised direction, training duration of the method, effect of backbone architectures and performance of the method regarding the number of labeled target samples.
3. Showcases the better usage of self-supervised consistency learning in comparison with adversarial training used in most domain adaptation methods.

Weaknesses:
1.  Lack of novel ideation for the problem statement. The algorithm seems to be inspired from a combination of multiple methods in conjuncture like Semi-supervised learning (FixMatch) and Co-training and applying the concept on multiple model outputs to achieve Inter-CR and Intra-CR adaptation stages in the SSDA problem.
2. Lack of proper reporting of results for the main experimental results. The results for DomainNet dataset using AlexNet backbone compares with CLDA and misses out on comparing with DeCOTA[1] and MCL[4] while the results using ResNet34 backbone does not compare with them. Moreover there are related works which should have been used in the comparison of the results like DeCOTA[1], ATDOC[2], STAR[3] and MCL[4]. Resnet34 backbone results can also include the 1-shot results. Similarly results for Office-31 and Office-Home are evaluated on different backbones of AlexNet and VGG respectively. More results on different backbones would be required for better comparison.
3.  Comparison to FixMatch is essential since the consistency regularization component is inspired from FixMatch.
4. Interesting observation is that the method performs poorly when the source and the target domains are comparable to each other. While the method seems to work well when the shift between both the domains are significant, it should also have the capability to do so otherwise.

Related works:
1. Yang L, Wang Y, Gao M, Shrivastava A, Weinberger KQ, Chao WL, Lim SN. Deep co-training with task decomposition for semi-supervised domain adaptation. In Proceedings of the IEEE/CVF International Conference on Computer Vision 2021.
2. Liang J., Hu D., Feng J. Domain adaptation with auxiliary target domain-oriented classifier IEEE conference on computer vision and pattern recognition (CVPR) (2021)
3. Improving semi-supervised domain adaptation using effective target selection and semantics. CVPRW 2021.
4. Multi-level consistency learning for semi-supervised domain adaptation. IJCAI 2022.

---

> ### Author Response · Authors · 2023-02-21
> **Reply for the comments -- part 1**
>
> Firstly, we are sorry for the late reply as we are busy recently. Secondly, we would like to thank the reviewers for their valuable comments. We reply to their comments below:
>
> 1. Lack of novel ideation for the problem statement. The algorithm seems to be inspired from a combination of multiple methods in conjuncture like Semi-supervised learning (FixMatch) and Co-training and applying the concept on multiple model outputs to achieve Inter-CR and Intra-CR adaptation stages in the SSDA problem.
>
> The novelty of this work comes from the utilization of complementary information by consistency regularization and integrating the learning objectives into a consistency regularization unified framework, which is easy to implement. The learning strategy of our work is inspired by Fixmatch, but the goal is different. Fixmatch is used for improving the performance of supervised learning, while the goal of our method is to transfer knowledge from one domain to the other domain. We will add the comparison in the revision.
>
>
> 2. Lack of proper reporting of results for the main experimental results. The results for DomainNet dataset using AlexNet backbone compares with CLDA and miss out on comparing with DeCOTA[1] and MCL[4] while the results using the ResNet34 backbone do not compare with them. Moreover, there are related works that should have been used in the comparison of the results like DeCOTA[1], ATDOC[2], STAR[3], and MCL[4]. Resnet34 backbone results can also include the 1-shot results. Similarly results for Office-31 and Office-Home are evaluated on different backbones of AlexNet and VGG respectively. More results on different backbones would be required for better comparison.
>
> Thanks for the advice. We mainly add the comparison with existing methods using different backbones on the DomainNet dataset.
>
> We add the comparison using the Alexnet backbone with MCL and don't compare with DECOTA as it doesn't provide the pre-trained model, which is needed for initialization. The results of the 3-shot using Alexnet on the DomainNet dataset are shown in this Table.
> |Method|R$\rightarrow$C| R$\rightarrow$P | P$\rightarrow$C| C$\rightarrow$S |S$\rightarrow$P |R$\rightarrow$S |P$\rightarrow$R |Mean|
> |----|----|----|----|-----|----|----|----|----|
> S+T|47.1| 45.0 |44.9 |36.4 |38.4 |33.3 |58.7 |43.4|
> MCL|55.4|   56.6    |52.4   |45.8   |48.6   |41.5   |**66.6**   |52.4
> BVCR | **62.7**| **58.4**| **56.5**| **50.3** |**49.6**| **51.8**|  62.8 | **56.0**|
>
>
> The results of 1 shot using resnet34 on DomainNet dataset is shown in this Table.
>
> |Method|R$\rightarrow$C| R$\rightarrow$P | P$\rightarrow$C| C$\rightarrow$S |S$\rightarrow$P |R$\rightarrow$S |P$\rightarrow$R |Mean|
> |----|----|----|----|-----|----|----|----|----|
> S+T |58.1 |61.8| 57.7 |51.5 |55.4 |49.1 |73.1 |58.1|
> DECOTA |**79.1** | **74.9** |**76.9**| 65.1| 72.0 |69.7 |79.6 |73.9|
> MCL| 77.4 | 74.6 | 75.5 | **66.4** | **74.0** | **70.7** | **82.0** | **74.4**|
> BVCR| 73.7| 73.3 | 69.9 | 65.1| 69.9| 66.8|75.5|70.6|
>
>
> The average accuracy on the Domainnet dataset with AlexNet backbone under 3-shot is 56.0%, which is higher than MCL. The average accuracy on the Domainnet dataset with Resnet34 backbone under 1-shot is 70.6%, which is lower than MCL and DECOTA. We do not achieve the best results in all settings, but one could not directly determine which method is better. More importantly, we don't want to surpass all related methods to achieve SOTA results. We would like to show an insightful method for the community, where dual/complementary views and implementation with unified consistency regularization are the most valuable.

---

> > ### Author Response · Authors · 2023-02-21
> > **Reply for the comments -- part 2**
> >
> >
> > 3. Comparison to FixMatch is essential since the consistency regularization component is inspired from FixMatch.
> >
> > Thanks for the advice. We implement Fixmatch on the Office-Home dataset with VGG backbone under 3-shot. The following is the result.
> >
> > |Method|R$\rightarrow$C| R$\rightarrow$P | R$\rightarrow$A| P$\rightarrow$R |P$\rightarrow$C |P$\rightarrow$A |A$\rightarrow$P |A$\rightarrow$C| A$\rightarrow$R |C$\rightarrow$R |C$\rightarrow$A| C$\rightarrow$P |Mean|
> > |----|----|----|----|-----|----|----|----|----|----|----|----|-----|----|
> > S+T| 49.6 |78.6 |63.6| 72.7 |47.2 |55.9 |69.4| 47.5 |73.4 |69.7 |56.2| 70.4| 62.9|
> > Fixmatch|52.3|78.7|63.7|72.9|47.7|54.9|70.5|49.7|72.8|69.4|57.7|70.0 |63.4|
> > BVCR |**63.2**|**84.4**|**69.1**|**76.7**|**60.9**|**61.7**|**79.0**|**57.4**|**76.1**|**73.4**|**62.4**|**79.4**|**70.3**|
> >
> > As we can see, Fixmatch achieves better performance than S+T but the improvement is limited, i.e., 0.5\%, which implies that simply treating a semi-supervised domain adaptation(SSDA) problem as a semi-supervised learning (SSL) problem is a trivial solution as this strategy could not deal with the domain distribution shifts effectively. BCVR considers the SSDA problem from a complementary view, i.e., intra-domain supervised information and inter-domain supervised information, which could better deal with the domain discrepancy and promote model learning.
> >
> > 4. Interesting observation is that the method performs poorly when the source and the target domains are comparable to each other. While the method seems to work well when the shift between both the domains are significant, it should also have the capability to do so otherwise.
> >
> > The motivation behind our work is to utilize complementary information for better learning. So as observed, the larger the domain discrepancy, the better the complementary information is, and the proposed method is expected to work better.

---

### Review · Reviewer_MqnC · 2023-01-11

**Summary Of Contributions:**

The paper gives a semi-supervised domain adaptation method called BVCR. The setting considered has a large amount of labeled data in the source domain and small amount of labeled data in the target domain. The goal is to learn a model that has good performance on the unlabeled pool in the target domain.  Their method trains two models separately ( one for each domain). In addition to usual classification loss on the labeled samples, their method also adds consistency regularization to minimize intra-domain and inter-domain discrepancies that help in getting a better adapted model. For consistency regularization they utilize weak and strong augmentation from (Sohn et al. 2020). The proposed method is backed by empirical evaluation on benchmark datasets and compared against standard baselines.


**Audience:**

Yes

**Claims And Evidence:**

Yes

**Requested Changes:**

1. Please provide error bars in all the experiments.

2. More intuition and details on the loss functions will be helpful.

3. One of the main pitches of the paper is that “target supervised information gets overwhelmed by source supervised information and the proposed method somehow address this issue” Can this claim be demonstrated more directly using some other evaluation metrics? An experiment in the earlier sections of the paper highlighting this problem with current methods can make the motivation well-grounded.

4. Please address writing related concerns listed above.

**Strengths And Weaknesses:**


Strengths:
1. Overall the paper is well written except for some minor issues ( listed below).
2. The proposed method brings together interesting ideas like consistency regularization using weak and strong augmentation, training two models separately one for each domain, to address various challenges in semi-supervised domain adaptation.
3. The method is empirically evaluated on several standard domain adaptation tasks and the results show that it achieves superior target domain accuracy in comparison to various baselines. Ablation studies justify the use of three losses viz. classification loss, inter and intra consistency regularization losses.

Weaknesses:
1. I did not get a good understanding of the loss functions and how exactly are they helping? More intuition and details in section 3.1, 3.2 and 3.3 could be helpful.
2. What is the interaction among the losses? In particular the modeling scheme is supposed to help in avoiding the target supervised information being overwhelmed by source supervised information, do other losses defeat this purpose?
3. The paper lacks novelty, the main ideas are from (Sohn et al. 2020).
4. No error bars in the experiments makes it hard to judge the effectiveness of the method.
5. Writing related :
    a) $s$, $t$ in the superscript are used for source and target domain. Using superscript $s$ with $u_i$  to denote strong augmentation is a bit confusing in the beginning.
    b) I found the use of loose cumbersome phrases (without formal/informal definitions) a bit uneasy to follow, especially in the abstract before I read the whole paper. e.g. “information interaction bidirectionally” , “source/target supervised information” , “inter/intra domain consistency regularization”.

---

> ### Author Response · Authors · 2023-02-21
> **Reply for the comment**
>
>
>
> Firstly, we are sorry for the late reply as we are busy recently. Secondly, we would like to thank the reviewers for their valuable comments. We reply to their comments below:
>
> 1. Please provide error bars in all the experiments.
>
> Thanks for the advice. Error bars are used on graphs to indicate the error or uncertainty in a reported measurement. They give a general idea of how precise a measurement is, or conversely, how far from the reported value the true value might be. While it is time-costing to add error bars in all experiments. Besides, previous methods such as DECOTA, MCL, and CLDA also don't provide error bars. So we will add the error bars in important experiments such as the Office-Home dataset.
>
> 2. More intuition and details on the loss functions will be helpful.
>
> To sum up, we find the supervised information from the source domain and the supervised information from the target domain is complementary to the unlabeled target samples. So we model the supervised information separately (3.1), explore the information within each model(3.2), and exchange the supervised information from two models for better model learning(3.3). In section 3.1, by learning a separate model for each domain on supervised samples, we could avoid target supervised information getting overwhelmed by source supervised information. In section 3.2, before exchanging the information from these two models, we use the unlabeled target samples to adapt the models, such that the source model could concentrate more on the domain-shared information(features) and less on source-private information(features), and the target model could learn better representations by utilizing the structural information within the unlabeled target samples.  In section3.3, we exchange complementary information for better learning. We will add the explanation in the revision.
>
>
> 3. One of the main pitches of the paper is that “target supervised information gets overwhelmed by source supervised information and the proposed method somehow address this issue” Can this claim be demonstrated more directly using some other evaluation metrics? An experiment in the earlier sections of the paper highlighting this problem with current methods can make the motivation well-grounded.
>
> Thanks for the advice. what we want to claim is that as the number of labeled target samples is much more than that of labeled source samples, so the learned model with these two samples would be biased to the source domain. I think the evaluation metrics could be the accuracy of the target samples, and we could compare the accuracy before and after adaptation, which is duplicated with the result in the section Experiment, so we don't add such experiments.
>
> 4. Please address writing related concerns listed above.
>
> Thanks for the advice, we will polish our paper to reduce the confusing symbols and phrases in revision.

---

### Decision · Action_Editors · 2023-02-17

**Recommendation:** Accept as is

**Comment:**

The authors work in the area of semi-supervised domain adaptation, where the goal is to perform domain adaptation with only a small number of labeled datapoints from the target domain. This is a hard problem in general, with a lot of moving parts. The authors put together several approaches that are often useful in semi-supervised learning in general. These include a mix of consistency regularization (for obtaining useful information from unlabeled data), both within and across domains. The outcome is a large pipeline that tries to get the most out of the available information.

The authors perform a substantial number of experiments, with reasonable performance improvements in a number of settings. The claims in the paper are substantiated to a solid degree.

All reviewers agree that there is some creativity in the way the authors brought multiple techniques, and that the experimental results are strong. Several reviewers liked the writing and presentation.

On the downside, the most common concern for reviewers is novelty. However, these concerns are not warranted for TMLR. Once these worries are removed from account, I believe the paper is in good shape.

For this reason, I believe the paper should be accepted. The authors are encouraged to clarify in writing any remaining points that tripped up some reviewers in terms of writing.

**Audience:**

Yes, this paper is in an important area for the audience, domain adaptation. Specifically, the authors work with semi-supervised settings, where there is a small amount of labeled datapoints for the target domain.

**Claims And Evidence:**

Yes, there is solid evidence presented for the authors' claims.